# A Deep Learning Ensemble for Network Anomaly and Cyber-Attack Detection

**DOI:** 10.3390/s20164583

**Published:** 2020-08-15

**Authors:** Vibekananda Dutta, Michał Choraś, Marek Pawlicki, Rafał Kozik

**Affiliations:** Institute of Telecommunications and Computer Science, UTP University of Science and Technology, Kaliskiego 7, 85-976 Bydgoszcz, Poland; chorasm@utp.edu.pl (M.C.); marek.pawlicki@utp.edu.pl (M.P.); rafal.kozik@utp.edu.pl (R.K.)

**Keywords:** anomaly detection, cyber-attacks, data pre-processing, deep learning, feature engineering, machine learning, network intrusion

## Abstract

Currently, expert systems and applied machine learning algorithms are widely used to automate network intrusion detection. In critical infrastructure applications of communication technologies, the interaction among various industrial control systems and the Internet environment intrinsic to the IoT technology makes them susceptible to cyber-attacks. Given the existence of the enormous network traffic in critical Cyber-Physical Systems (CPSs), traditional methods of machine learning implemented in network anomaly detection are inefficient. Therefore, recently developed machine learning techniques, with the emphasis on deep learning, are finding their successful implementations in the detection and classification of anomalies at both the network and host levels. This paper presents an ensemble method that leverages deep models such as the Deep Neural Network (DNN) and Long Short-Term Memory (LSTM) and a meta-classifier (i.e., logistic regression) following the principle of stacked generalization. To enhance the capabilities of the proposed approach, the method utilizes a two-step process for the apprehension of network anomalies. In the first stage, data pre-processing, a Deep Sparse AutoEncoder (DSAE) is employed for the feature engineering problem. In the second phase, a stacking ensemble learning approach is utilized for classification. The efficiency of the method disclosed in this work is tested on heterogeneous datasets, including data gathered in the IoT environment, namely IoT-23, LITNET-2020, and NetML-2020. The results of the evaluation of the proposed approach are discussed. Statistical significance is tested and compared to the state-of-the-art approaches in network anomaly detection.

## 1. Introduction

Critical structures such as Internet Industrial Control Systems (ICS) and Sensitive Industrial Plants and Sites (SIPS) need to be functional and operate reliably even when subjected to unforeseen threats or external attacks. A number of systems are dedicated to mitigating the cascading impact resulting from the failure of or an attack on critical infrastructure. Those systems, however, can be the subject of a cyber-attack [1].

The attack surface of SIPS is wide and must be protected in both cyber and physical spaces. It comprises the SIPS data management, control, and communication layers. The main goal of a malicious user is to get access to one of these layers to steal or tamper with sensitive data. This is doable physically, remotely, or through a combined vector of attack. On the transmission side, SIPS are exposed to cyber-attacks. The likely targets of the cyber-attacks are remotely controlled devices (e.g., switches, transformer taps, valves). Their hijacking could potentially damage several physical assets and cause widespread losses. A likely target that the malicious user can attack is the server machines where sensitive data are stored. Finally, the attacker could modify critical measurements reported to the operator for controlling or monitoring the infrastructure units.

Opposing malicious software [2] that could be launched against one of the attack surface layers constitutes one of the most significant cybersecurity demands. Malicious software in computer programs frequently stays passive during the scanning of infected infrastructures and assets. Some infected systems have the ability to connect with other infected machines to establish a botnet, allowing cybercriminals a range of transgressions, including DDoS attacks, spam, ransomware, and subtle data theft. The rapid development of novel technologies, like cloud computing, the Internet of Things (IoT), and NB-IoT, comes with new vulnerabilities, and they dangerously increase the lack of trust concerning cybersecurity, which many end-users already exhibit [3].

Network intrusion detection methods have progressed from mechanisms relying on port inspection to techniques making full use of machine learning. The common port-based approaches have become outdated since current applications mostly rely on dynamic port allocation rather than on registered port numbers. The increase in the volume of encrypted traffic causes the payload-based approaches to be unsuccessful. This leads the cybersecurity specialists in the direction of utilizing machine learning and network flow features. Recent advancements in machine learning methods for network anomaly detection have been warmly welcomed [4]. Due to the heterogeneous and diverse nature of cloud environments, machine learning provides the answers to the challenges induced due to the existence of virtualized environments with their wide range of application workloads [5].

Some of the contemporary solutions use a correlation engine with the Kafka architecture that performs effective anomaly-based intrusion detection for streaming data in the application layer. This kind of system adopts the lambda architecture featuring the scalable data processing framework named Apache Kafka, which facilitates an engine that processes Big Data workloads. The key aspects of this kind of design enable a distributed computing system that integrates multiple machine learning and deep learning models. This particular example also facilitates an application module that offers different intrusion detection tools and a visualization module for the end-users.

### 1.1. Motivation and Objectives

Intrusion Detection Systems (IDSs), working as remote components or as part of a larger Intrusion Detection Network (IDN), are designed to facilitate attack detection capabilities to protect a network. This paper considers the problem of network anomalies, providing a machine learning-based anomaly detector and optimal solutions to make the detection tools resilient.

The objective is to improve the classification effectiveness of intrusion detection systems and networks operating in adversarial settings by developing an ensemble method facilitated by stacked generalization techniques to analyze the outliers. Diverse anomaly detection methods may cause different results on the same data. Thus, the task of offering a suitable technique for a specific domain is rather challenging. The stacked generalization approach is utilized to enhance model performance [6]. The scope of this work covers the following objectives:Selection of the latest network traffic datasets featuring labeled network flows and corresponding features to evaluate the proposed approach;developing a suitable model that integrates the key features in the feature engineering phase to facilitate the design architectures that resist, at least to some extent, certain types of deliberate attempts to evade detection or, more generally, subvert the protection offered by the proposed method;the design of the deep models integrated by the stacked generalization method, which is generated by a group of selected models (designated by the ensemble) followed by a meta-classifier that learns the best way of facilitating the synergy of a group of models.

### 1.2. Contributions and Organization

In this work, the problem of an efficient anomaly-based network intrusion detection method is discussed. It involves a careful examination of recent benchmark datasets, IoT-23, LITNET-2020, and NetML-2020, with promising accuracy and minimal computational complexity. The work contained in this text aims at providing an efficient anomaly detection mechanism meeting the following description:featuring flow-level analysis utilizing novel methods of aggregation and class balancing using the Synthetic Minority Over-sampling Technique (SMOTE) followed by the Edited Nearest Neighbors (ENN) approach to enhance the performance of the classification efficiency via extracting only significant features with the capability of coping with network data on a large scale;developing and applying a dimensionality reduction approach such as a Deep Sparse AutoEncoder (DSAE) for minimizing the size of the input feature vector to the classifier in order to reduce the computational complexity;a stacked ensemble approach using different Level-0 models (such as DNN, LSTM) and a Level-1 model (Logistic regression) is offered, and its performance is evaluated on three benchmark datasets;the disclosed set of techniques is thoroughly evaluated through a series of experiments to illustrate the extent of the improvements in the detection rate in the chosen datasets in comparison to several state-of-the-art techniques.

The paper discusses this contribution in detail. In Section 2, the related works are reviewed. Section 3 gives an introduction about the anomaly-based intrusion detection platform, its data pre-processing approach, feature engineering techniques, and the proposed methodology of the stacked ensemble machine learning approach. Section 4 introduces the recently built IoT-23, LITNET-2020, and NetML-2020 flow-based datasets and their characteristics. Section 5 discusses the experimental results. Finally, in Section 6, conclusions are drawn, and the scope of future research is elaborated.

## 2. Related Work

The significance of the subject and the focus on network traffic analytics brought multiple recently-published studies. In this section, various works on the use of machine learning to detect anomalies in computer networks are investigated.

### 2.1. Deep Learning

When it comes to machine learning techniques, it is especially important to take note of deep learning and Deep Neural Networks (DNN). With the emergence of high-computational GPUs, these have revolutionized artificial intelligence and many areas of engineering and computer science. It should be expected that it will be similar in the area of cybersecurity. Although the techniques utilizing DNNs in IDS are still in their early stages, there are several examples of their usage in cybersecurity. In their work, Jihyun et al. [7] utilized Long Short-Term Memory (LSTM) networks and used the KDDCup ’99 dataset. The applied LSTM-RNN achieved 96.93% accuracy and a recall of 98.88% [8]. Abolhasanzadeh [9] used a deep autoencoder as an attack detector in Big Data. The conducted experiment on the NSL-KDD test set was applied to achieve the bottleneck features in dimensionality reduction as part of intrusion detection. It was concluded that the achieved accuracy was promising for real-world intrusion detection [8]. In [10], Fiore pondered the utilization of a Restricted Boltzmann Machine (RBM). The network would be fit for anomaly detection by allowing it to learn on a real-world dataset collected from 24 h of workstation traffic [8]. Alom et al. [11] offered the Deep Belief Network’s (DBN) abilities to perform intrusion detection through a series of experiments. The authors trained the DBN on NSL-KDD data to distinguish unforeseen attacks. Further development of hardware resources and computing capabilities of GPU cards makes DNN methods an interesting alternative to existing traditional methods, and they can be seen as a panacea to overcoming the challenges, which are faced by institutions responsible for CIP.

### 2.2. Ensemble Learning and Stacked Generalization

The literature on machine learning proposes multiple approaches regarding ensemble designing. Ensemble learning is an approach that builds on top of a collection of models. The method very often performs better than any of those models could on their own. This learning paradigm found its applications in both supervised learning and in unsupervised approaches [12]. An ensemble design involving the Level-0 and Level-1 model demonstrates superior outcomes by consolidating different individual models. Govindarajan et al. [13] introduced a hybrid model comparing individual models in a classification task. The authors of this work indicated that building an ensemble improves the accuracy of the model tremendously in a task involving both normal and abnormal traces from a mail application. Hansen et al. [14] found in their research that for an ensemble model to be more accurate as compared to the individual models, (a) every single model has to perform better than random guessing, and (b) singular models need to have output uncorrelated with one another (the ensemble has to be manifold).

Another form of increasing the performance of machine learning models can be achieved by stacking them. In this approach, also known as stacking generalization, the prediction outcomes of the models are combined using a meta-classifier [15]. Stacked Generalization (SG) was initially introduced by Wolpert [16]. Following the three-step formula, stacking is accomplished when: (a) the creation of models is accomplished by utilizing at least two different learning methods; (b) the outputs of the models contribute to the creation of a novel dataset in conjunction with the original hold-out set; (c) this novel dataset is used to create an all-encompassing model taking into account the preceding outputs [17]. Cerqueira et al. [18] investigated a method similar to stacking, where several models, e.g., LOFand hierarchical agglomerative clustering, provide the prediction outcomes, which are appended to the original dataset. In [19], Karthick et al. merged the hidden Markov and naive Bayesian models in order to facilitate flexible intrusion detection. The research suggested that this stacked generalization provides satisfactory results.

The literature review suggests that flow-based data methods in network intrusion detection are winning the attention of researchers. There is always a high demand for novel cybersecurity datasets, and the question of an ideal set of network flow features still remains open.

## 3. Proposed Methodology

The most significant objective of this research is to obtain a reliable classification of outliers by using a stacked ensemble method. In the following section, a general synopsis of the framework is offered. Then, the data pre-processing, feature engineering, and classifier modeling steps are discussed in detail.

### 3.1. General Overview

This section presents an overview of the proposed deep-stacked ensemble approach applied in anomaly detection on network traffic data. The detailed architectural diagram is depicted in Figure 1.

The individual phases of the proposed framework are (1) dataset selection, (2) data preprocessing integration by feature engineering, data balancing, and dimensionality reduction using a deep sparse autoencoder followed by (3) data output, (4) data splitting, and (5) classification into “normal/anomaly” using a stacked ensemble approach unified by deep models and a meta learner (i.e., logistic regression). The detailed descriptions of those steps are provided in Section 3.2, Section 3.3, Section 3.4 and Section 3.5, respectively.

### 3.2. Data Pre-Processing

Fairly frequently, the methods of data collection produce attributes that are either duplicated or superfluous for network data obtained via analysis of network traffic [20]. Clearing the insignificant and unnecessary information is a step to produce a more robust representation, which supplies better applicable inputs to the classifier [21]. The preprocessing steps of our approach are provided below.

#### 3.2.1. Feature Selection

NetFlow datasets often include non-identical feature attributes that are categorized as flow, basic, content, time, additionally generated, and labeled features, respectively. However, the information gathered from packet captures also provides a variety of irrelevant or redundant details. Removing unnecessary information offers better, unbiased detection. In addition to the above pre-processing, the selected datasets (i.e., IoT-23, LITNET-2020, NetML-2020) contain a certain amount of unusable values. Feature imputation was performed to replace the infinities with max values. For missing records, the feature means were used.

#### 3.2.2. Feature Normalization

The continuous values in the network traffic datasets have a variety of magnitudes. This causes problems for a variety of classifiers. Thus, scaling is performed as a means of normalizing the attributes, squeezing the values to the range of 0 to 1. Therefore, the features are scaled according to Equation (Equation 1):(1)$${\bar{x}}_{m,n}=\frac{x_{m,n}}{\max_{n}\left(x_{m,n}\right)},\;\;\forall_{m=1,\ldots ,f},\;\forall_{n=1,\ldots ,i},$$
where ${\bar{x}}_{m,n}$ represents the scaled feature, $\max_{n}\left(x_{m,n}\right)$ is the maximum value of the data in the *m*-th feature, *i* is the number of samples in both the training and testing data, and *f* is the number of features from the feature selection process.

### 3.3. Data Balancing

In situations where the learning dataset has an imbalance in class distribution, machine learning algorithms may experience problems. In imbalanced data learning, under-sampling the majority class is a conventional plan of operation. Oversampling techniques followed by undersampling can be used to balance the datasets. We used two established approaches, namely the Synthetic Minority Over-sampling Technique (SMOTE) and Edited Nearest Neighbors (ENN), to balance the IoT-23, LITNET-2020, and NetML-2020 datasets [22,23]. Recently, hybrid approaches have become popular [24]. Methods like SMOTE+ENN, among other, have often been utilized for alleviating the issue of class imbalance to boost the efficiency of the classifier.

The capability of the selected method, in which the training datasets were cleaned or reduced, is illustrated in Section 3.2.1 and Section 3.2.2. The count of samples in each class in the pre-processed dataset is subjected to the balancing procedure. Following [24], the SMOTE+ENN approach for class balancing is presented in Algorithm 1.
**Algorithm 1** SMOTE+ENN.**Input:** Training dataset tr;
**Output:**
$tr^{\prime }$
 **Initialization**:
 1: $tr^{\prime }$ = DB (tr)
 **End**
 *DB (tr)*:
 2: $tr_{new}$ = SMOTE (tr) // Over-sampling the training set
 3: $tr_{tmp}$ = ENN ($tr_{new}$) //Remove noise sample
 4: **return**
$tr_{tmp}$


After applying the SMOTE+ENN method to enhance the pre-processed data (see Section 3.2), the artificially balanced dataset is utilized as a training set for the neural network learning process. It is worthwhile to mention that the term tr represents ($X_{tr}$, $y_{tr}$).

### 3.4. Feature Dimensionality Reduction

The DSAE learning algorithm is employed for dimensionality reduction. The autoencoder can learn the latent representation of features in a reduced space in an unsupervised setting. The DSAE builds on the concept of an autoencoder adding the sparse penalty term, which hinders feature learning to achieve a concise representation of the input vector [25,26]. Furthermore, an autoencoder using non-linear activation functions and multiple layers possesses the ability to obtain non-linear relationships, unlike Principal Component Analysis (PCA). It is also more efficient, in terms of model parameters, to train several layers of an autoencoder rather than train one huge transformation with PCA.

Following [25], the input vector *x* defines sets of NetFlow data $x=\{x_{1},x_{2},\ldots .,x_{n}\}$, which are later reconstructed into an N×M dataset; therefore, $x\left(i\right)\in R^{M}$. These NetFlow data are used as an input matrix *X* initially condensed to a lower dimension that is expressed by a set of one or more hidden layers $h=\{h_{1},h_{2},\ldots ,h_{n}\}$. The hidden representation of $\bar{h}$ is then depicted as the output $\bar{x}=\left\{{\bar{x}}_{1},{\bar{x}}_{2},....,{\bar{x}}_{n}\right\}$, respectively.

Let us consider *k* as the token node parameter in the nodes of layer *l*, and *j* is the counter parameter in the preceding layer l−1. Therefore, the outcome of the neurons in a computational layer *h* can be expressed by Equation (Equation 2).
(2)$$h_{k}^{l}=f\left(\sum_{j=1}^{n}W_{kj}^{\left(l-1\right)}\cdot h_{j}^{\left(l-1\right)}+b_{k}^{\left(l-1\right)}\right)$$

The size of the weight matrix is expressed by $W\in R^{m\times n}$, and the bias is $b\in R^{m}$. Instead of *ReLU* or *tanh*, *sigmoid* is utilized for neural activations.

The applied reconstruction error $E_{r}$ between the input vector *x* and the reconstructed input $\bar{x}$ followed by the mean squared error function is shown in Equation (Equation 3).
(3)$$E_{r}=\frac{1}{n}\sum_{i=1}^{n}x_{i}+{\bar{x_{i}}}^{2}$$

*n* represents the count of input examples. DSAE is chosen to obtain a reconstructed depiction of the input vector. Hence, sparsity is included in the DSAE cost function to cap the average activation value of neural nodes in the computational layers. Let ρ be the average activation value of sparse parameters, and a penalty term is employed to prevent $\rho_{k}$ from deviating from parameter ρ. Therefore, the mean activation value of the *k*-th neuron in the hidden layer is expressed by:(4)$$\rho_{k}=\frac{1}{n}\sum_{i=1}^{n}\left[h_{k}\left(x\left(i\right)\right)\right]$$
where both *n*’s represent the count of input data points and *i* the sample. Since the average activation $\rho_{k}$ approaches ρ (a constant close to 0), the KL divergence is adopted to facilitate sparsity.
(5)$$KL(\rho ||\rho_{k})=\rho \log\frac{\rho }{\rho_{k}}+\left(1-\rho \right)\log\frac{1-\rho }{1-\rho_{k}}$$

Next, the cost function of the neural network is set as $c(W,b,x,\bar{x})$. Adding the sparse penalty term is expressed by Equation (Equation 6).
(6)$$c_{sparse}=c\left(W,b,x,\bar{x}\right)+\beta \sum_{k=1}^{m}KL((\rho ||\rho_{k})$$

The β rate influences the strength of the sparsity penalty. L2 Regularization (L2R) is employed in the cost function as a means to avoid overfitting (see Equation (Equation 7)).
(7)$${\bar{c}}_{sparse}=c_{sparse}+{\alpha ||w||}^{2}$$
where α is a coefficient and ${\left||w|\right|}^{2}$ represents the penalty term. This means that an increase in the value of α translates to a deeper weight attenuation.

The work contained herein considers a three hidden layer sparse auto-encoder (so-called deep sparse autoencoder), which utilizes the sigmoid in place of the neural activation functions. The input layer has *N* nodes depending on the selected dataset: IoT-23, LITNET-2020, and NetML-2020. *N* varies taking into account the total number of features in each dataset after the data pre-processing phase. The initial computational layer reduces the dimensions to 19 (IoT-23), 30 (LITNET-2020), and 20 (NetML-2020) nodes with a favorable error approximation. The following layer decreases the number of features to 15 (IoT-23), 20 (LITNET-2020), and 16 (NetML-2020). Finally, in the third layer, the features are reduced to 10 (IoT-23), 15 (LITNET-2020), and 12 (NetML-2020), respectively. Once the learning stage of the autoencoder is successfully concluded, the network is placed as the input vector provider to the classifier in the final stage. The following set of parameters was determined by running preliminary experiments with a trial and error approach: the weight decay λ = 0.0003, the sparsity parameter ρ = 0.5, and the sparsity penalty term β = 6, respectively. The method is delineated in Algorithm 2, and the visualization of the applied dimensionality reduction on each dataset is presented in Figure 2.
**Algorithm 2** Deep Sparse AutoEncoder (DSAE).**Input:** Input vector *x*
**Output:** Reconstructed input $\bar{x}$
 **Initialization**:
 1: Initialized weights *W* and thresholds *b*
 2: Obtain the reconstruction error $E_{r}$ according to Equation (Equation 3)
 3: Add sparsity regularizer to cost function according to Equation (Equation 5)
 4: Add L2 regularization to cost function according to Equation (Equation 7)
 **End**


### 3.5. Classifier Modeling

The proposed framework utilizes a stacked generalization approach over flow-based data. The results obtained by the Level-0 classifiers ($\bar{x}$) are forwarded to the meta-classifier. Thus, the overall approach incorporates two different deep learning algorithms (Deep Neural Network (DNN), Long Short-Term Memory (LSTM)) and a meta-classifier (Logistic Regression) (illustrated both in Section 3.5.1 and Section 3.5.2). Stacked generalization is employed to synergize the deep learning methods, as a way to cooperatively use several models to potentially eliminate the generalizing biases concerning a particular training set. Figure 3 illustrates the concept, which consists of the base and meta-classifier. The implementation of a stacked generalization involves two kinds of models: (a) base models (Level-0 classifiers) and (b) meta-models (Level-1 or meta-classifier). The essence of stacked generalization lies in using the Level-1 classifier to perform the task by learning from Level-0 models.

In general, machine learning can be viewed as a minimization of the problem as shown in Equation (Equation 8).
(8)$$\min_{f}\sum_{i=1}^{n}l\left(f\left(x_{i}\right),y_{i}\right)+\lambda r\left(f\right)$$

The first term of Equation (Equation 8) is the loss function, which expresses the distance between the predicted and real values. The second term in Equation (Equation 8) is a regularization term that measures the complexity of the function *f*.

Furthermore, one of the key methods in the proposed approach is to obtain the training set ($d_{cv}$) for the Level-1 model ($m_{mtcf}$) using the cross-validation approach. For example, given a pre-processed dataset *d* randomly split into *K* equal folds {$d_{1}$, $d_{2}$,...,$d_{K}$} (in this work, *K* = 5), let us consider $d_{k}^{tr}$ and $d_{k}^{te}$ to be the training and test sets for the *k*-th fold in a *K*-fold cross-validation. Given different deep learning models ($M_{1}$, $M_{2}$,....,$M_{L}$), where L=2 in this research, each $M_{L}$ is trained by $d_{k}^{tr}$ and predicts each instance on $d_{k}^{te}$. Let $p_{k}$($\bar{x}$) denote the prediction of the model $M_{L}$ on $\bar{x}$. Then, we have:(9)$$z_{kn}=p_{k}\left({\bar{x}}_{n}\right)$$

At the end of the entire cross-validation process of each model $M_{L}$, the data ($d_{cv}$) are an ensemble from the outcome of the *L* models (see Equation (Equation 10)).
(10)$$d_{cv}=\left(y_{n},z_{1n},...,z_{Ln}\right),\;\;n=1,2,\ldots ,N$$
where $d_{cv}$ is the training set for the Level-1 model (i.e., meta-classifier) $m_{mtcf}$. Hence, to finish the training, Level-0 models $m_{L}$ are trained using the pre-processed dataset *d*, and $m_{mtcf}$ is trained by $d_{cv}$ followed by 5-fold cross-validation. Now, in the testing phase (i.e., hold-out set for testing), given a new instance, Level-1 model $m_{mtcf}$ produces the final outcomes for that instance. It is important to mention that 5-fold cross-validation was performed and obtained the average accuracy, standard deviation, and standard error of the mean as presented in Table 1, Table 2 and Table 3, respectively. The offered framework (Figure 3) works via the end-to-end prototype where the Level-1 classifier (i.e., meta-classifier) is trained by the $d_{cv}$ obtained from the Level-0 classifier.

#### 3.5.1. Modeling the Deep Neural Network

Following [27], the structure of the DNN utilized in the proposed stacking framework for an anomaly detection system is described below.

We initialize a four layer neural network. Back-propagation is used to train the network with stochastic gradient descent as the optimizer. The input data are propagated through the hidden layers and then transformed to the final output. A loss function is employed to penalize the network by back-propagating the error to adjust the weights. The network parameters (i.e., weights) are updated for each mini-batch at each iteration (i.e., epoch).

The proposed model has four subsequent layers with 52, 34, 26, and 16 nodes on the hidden layers. The last part of the model is a fully connected neural network with 2 output nodes. Furthermore, the following parameters of DNN are defined as the activation function (ReLU, Sigmoid), optimizer (Adam), batch size (512), epochs (500), and loss function (binary–crossentropy). This model is also using an early stopping algorithm to gain the best value of validation accuracy.

#### 3.5.2. Modeling Long Short-Term Memory

To obtain the time-related non-linear dynamics in the provided data, an LSTM network is utilized. LSTM is an improvement over the RNN architecture, extending the ability to capture long-term dependencies [28].

A four layer LSTM is built on top of the DSAE to obtain the encoded temporal patterns. Finally, a fully connected dense layer is stacked together to process the outputs of the LSTM. The utilized optimizer was *Adam*. The activation functions (tanh, Sigmoid), batch size (512), epochs (500), and loss function (binary-–crossentropy) were also defined as the network parameters. Similar to DNN, this model also uses the early stopping algorithm to gain the best value of the validation accuracy.

## 4. Selection of Datasets

Labeled network traffic datasets facilitate the power of supervised methods to offer necessary information to train the IDS efficiently in order to achieve extraordinary accuracy and dependability in distinguishing a wide range of network attacks.

Recently, several new flow-based benchmark datasets have been introduced to the public domain such as IoT-23, LITNET-2020, and NetML-2020. Due to their temporal proximity, these have not yet been widely used by the cybersecurity community. However, we use these datasets for efficient anomaly-based network intrusion detection, on the realistic and up-to-date network traffic data.

### 4.1. IoT-23

IoT-23 (https://www.stratosphereips.org/datasets-iot23) is a network traffic dataset that incorporates 20 malware subsets and three benign subsets. The dataset was initially made available in January 2020 by the Stratosphere Laboratory in Czechia. The objective of the dataset is to present a sizable dataset of labeled malware and benign traffic coming from real captures to develop intrusion detection tools featuring machine learning algorithms.

The following labels in all 20 malicious captures are: Part-Of-A-Horizontal-PortScan (213,852,924 flows), Okiru (47,381,241 flows), Okiru-Attack (13,609,479 flows), DDoS (19,538,713 flows), C&C-Heart Beat (33,673 flows), C&C (21,995 flows), Attack (9398 flows), C&C (888 flows), C&C-Heart Beat Attack (883 flows), C&C-File download (53 flows), C&C-Tori (30 flows), File download (18 flows), C&C-Heart Beat File Download (11 flows), Part-Of-A-Horizontal-PortScan Attack (5 flows), C&C-Mirai (2 flows). On the other hand, the number of flows belonging to benign is 30,858,735 flows. Nevertheless, the dataset has 21 feature attributes including the class label. Therefore, a total of 21 attributes determining the features of connections are present in each data instance. The attributes are mixed in nature, with some being nominal, some being numeric, and some taking on time-stamp values. The general overview of the dataset is presented in Table 4, followed by a description of the attack class labels. Due to the large size of the IoT-23 dataset, only seven scenarios (Malware-1-1, Malware-3-1, Honeypot-4-1, Honeypot-5-1, Honeypot-7-1, Malware-34-1, Malware-43-1) were evaluated in this work. The feature attributes with their description in the IoT-23 dataset are presented in Table 5.

### 4.2. LITNET-2020

The LITNET-2020 (https://dataset.litnet.lt) NetFlow dataset [29] consists of senders and collectors. The senders are made up of Cisco routers and Fortige (FG-1500D) firewalls, which were utilized to evaluate NetFlow data passing through the collectors. The collector incorporates software that accounts for receiving, storing, and filtering data. The specific counts of data samples in the classes of the dataset (with a total of 45,492,310 flows) is presented in Table 6. All instances are categorized into ordinary data (45,330,333 flows) and malicious data (5,328,934 flows). The malicious instances are further categorized into nine classes, taking into account the type of network attack.

The data preprocessor selects initially 49 features that are specified to the NetFlow V9 protocol [30] to arrange the dataset. Furthermore, an additional 15 feature attributes are supplemented by the data extender. Having said that, an additional 19 attributes are offered to recognize attack types. Therefore, the final datasets have a set of 84 feature attributes. The feature attributes with their descriptions of the LITNET-2020 dataset are summarized in Table 7. The captures were accumulated in a real-world network.

### 4.3. NetML-2020

The NetML-2020 (https://evalai.cloudcv.org/web/challenges/challenge-page/526/overview) dataset was created for anomaly detection tasks by obtaining 30 traffic data from Stratosphere IPS (https://www.stratosphereips.org/). The flow features are extracted in the JavaScript Object Notation format by offering a raw *pcap* file as an input to the feature extraction tool, and each flow sample is listed in the output file. Finally, a unique id number is placed to identify every flow and the label information taking into account the raw traffic packet capture file. The NetML dataset provides 484,056 flows and 48 feature attributes (since we consider “top-level” granularity, only 26 meta-features were selected after the data pre-processing stage). The detailed description of the feature attributes and selected capture files is listed both in Table 8 and Table 9.

Every capture offers flows belonging to a different category. Flows derived from the capture—win1.pcap file are considered malicious in the highest granularity labels and *Tinba* in the bottom-level labels. Three annotations, “top-level”, “mid-level”, and “fine-grained”, are chosen to express the class granularity. At the highest granularity, the dataset is binary (benign or malware). The “mid-level” distinguishes specific software communication, e.g., Facebook, Skype, Hangouts, etc. Lastly, the “fine-grained” level classifies multiple distinct types of attacks, e.g., Ramit, HTBot, etc.

## 5. Experiments and Results

To assess the classification performance, metrics such as the accuracy score, precision, recall, F1 score, and Matthews Correlation Coefficient (MCC) were calculated. Additionally, geometric mean (g-mean), omnibus tests, and the Friedman rank were used as evaluation indicators to evaluate the algorithm’s results. It is important to mention that in this work, we consider binary classification: normal and malware (zero, one) for both individual algorithms and the proposed stacked ensemble framework, respectively.

Evaluation matrices and experimental settings are discussed both in Section 5.1 and Section 5.2, while Section 5.3, Section 5.4 and Section 5.5 present the result of the experiments (including statistical tests), respectively.

### 5.1. Evaluation Metrics

A standard suite of metrics was used in this work for the evaluation of the approach: Accuracy (ACC), Precision (Pr), Recall (Re), F1-score, False Positive Rate (FPR), Matthews Correlation Coefficient (MCC), and g-mean. The formulation of g-mean, as used in this work, is included in the next subsection.

#### Geometric Mean

This work uses the g-mean definition presented in [24]. The geometric mean (g−mean) is applied to assess the results given by the classifiers for datasets containing imbalanced classes; it is expressed in the following manner: (g−mean∈[0,1]):(11)$$g-mean=\sqrt[{M}]{\prod_{m=1}^{M}Acc_{m}}$$
where Acc signifies the accuracy of classification in a given category and *M* is the count of defined categories (in this case, *M* = 2).

In order to further analysis, a statistical test was performed, e.g., the *Friedman test* [31], a non-parametric method. The initial phase of this method performs a gradation of the algorithms for every selected dataset (i.e., best—1, second best—2, etc.).

### 5.2. Experimental Settings

The experiments were performed on a Linux setup with 16 GB RAM and the Intel Core i7 10-th Generation Processor. All the experiments were administered using the Python programming language and the TensorFlow/Keras/Scikit-learn stack.

### 5.3. Experimental Evaluation on the IoT-23 Dataset

The following section gives the evaluation of the results attained by the stacked ensemble classifier proposed against individual classifiers, namely DNN, LSTM, and random forest [27]. The IoT-23 dataset was used to assess the efficiency of the classifiers. It is one of the most recent network traffic datasets (20 malware, three benign), and the description is detailed in Section 3.1. Each classifier (i.e., DNN, LSTM, and stacked) underwent the learning procedure on the training set. The results are reported in terms of accuracy, average accuracy, standard deviation, and Standard Error of the Mean (SEM) using five-fold cross-validation, as shown in Table 1. These classifiers were trained and tested on the same data distribution. The performance comparisons are also reported using statistical tests and evaluation matrices such as the overall accuracy, g-mean score, and Friedman average rank.

Table 10 presents the overall accuracy and Friedman average rank, followed by the result of the g-mean score secured by the state-of-the-art classifiers vs. the stacked ensemble classifier. Note that a lower value in ranking represents improved classification results. Furthermore, the evaluation results taking into account the precision, recall, FPR, and MCC are presented in Table 11.

### 5.4. Experimental Evaluation on the LITNET-2020 Dataset

In this section, we performed our experiments with the most recent, publicly available benchmark dataset, LITNET-2020 [29], published by Kaunas Technological University in May 2020. To examine the performance of our framework using selected features, we first trained the individual classifiers and stacked ensemble framework using five-fold cross-validation and present the results in terms of accuracy, average accuracy, standard deviation, and SEM in Table 2.

Similar to Section 5.3, the detection performance of the tested methods such as DNN, LSTM, and random forest vs. stacked ensemble algorithms is also compared. The results are reported in terms of statistical tests and evaluation matrices such as the overall accuracy, F1-score, g-mean, and Friedman average rank in Table 12. The evaluation of the performance results concerning the precision, recall, false positive rate, and MCC is also presented in Table 13.

### 5.5. Experimental Evaluation on the NetML-2020 Dataset

In this section, we demonstrate the effectiveness and performance of the proposed stacked ensemble framework on the challenging NetML-2020 dataset [32]. This dataset was released during the open challenge Network Traffic Analytics using Machine Learning (NETAML) workshop sponsored by the Intel Corporation.

We selected two basic classification models including DNN and LSTM classifiers as our baselines since DNN is frequently utilized in multiple domains and LSTM presents a strong deep learning approach that can achieve the foremost increases with regards to accuracy. The selected baseline methods underwent the training procedure using the same data that were used by our proposed stacked ensemble framework. Five-fold cross-validation was performed, and the results are reported in Table 3.

Thereafter, the proposed stacked approach is compared with the baseline algorithms and the methods reported in [32], as shown in Table 14, and it showed an increase in performance over the baseline algorithms and those found in the subject literature. Similar to the previous section, the evaluation of the model performance with respect to precision, recall, FPR, and MCC is presented in Table 15.

The showcased performance increases prove a significant improvement in performance of the proposed stacked ensemble method as compared to the baseline. The method shows promise in terms of classification accuracy and other evaluation metrics. It was also investigated and proven experimentally that the deep sparse autoencoder improves the accuracy of the proposed algorithms as compared to the case where it was not employed [27,32]. The results also exhibit that the enhanced effectiveness can be obtained not only by improving the topology of the neural network or performing hyperparameter tuning, but also by enhancing the preprocessing stage.

Figure 4 depicts the duration of the training of the proposed framework comparing state-of-the-art methods. The training duration is obtained by calculating the computation time required for forming the classification model. During the investigation, the proposed classifier used in the final stage had appreciably lower training time as compared to more complex classifiers working with the inputs obtained as the output of Level-0 training ($d_{cv}$). It is to be noted that the baseline classifiers were also trained with the same set of outputs (including the algorithms previously used in the Level-0 classifier: DNN, LSTM).

Lastly, as a means to further show how effective the proposed method performs, we further analyze the cumulative amount of incorrect detection.

## 6. Conclusions and Future Work

This work addressed an ensemble approach incorporating deep learning algorithms using the concept of stacked generalization for an effective anomaly-based network intrusion detection system. In this paper, various feature engineering methods were applied together with dimensionality reduction to achieve the highest efficiency. A combination of DNN and LSTM followed by a meta-classifier resulted in significant performance and detection of anomalies w.r.t. the most recent network traffic datasets. Three heterogeneous datasets, IoT-23, LITNET-2020, and NetML-2020, were used to assess the effectiveness of the proposed stacked ensemble framework. Following statistical significance tests, we came to the verdict that the suggested approach outperforms the state-of-the-art individual classifiers and meta-classifiers such as random forest and support vector machine.

From the series of conducted experiments, it is inferred that the proposed approach provides a significant improvement in terms of evaluation metrics when validated against pre-specified testing sets. Briefly, the proposed framework can eliminate the challenge of providing recent network traffic datasets and provide an acceptable accuracy to detect anomaly behaviors in the desired network.

For future work, the implementation strategy can be further extended to conduct experiments on more sophisticated datasets if those can be acquired. Advanced computational methods like Apache Spark will be utilized in the future to boost the processing speed and facilitate the scalability for massive volumes of network data. Additionally, the approach is to be validated for solving a multi-class problem. At the moment, we also focus on the second part of the model (i.e., transfer learning). The study will apply a lifelong learning algorithm along with a deep learning one to make the method [33] more robust to unknown and known attacks. Finally, first steps have already been taken to secure the deep learning component itself against the threat of adversarial attacks [34], and we plan to continue research in that regard.

## 7. Data Availability

The datasets used in this work are publicly available for research purposes.

IoT-23: https://www.stratosphereips.org/datasets-iot23LITNET-2020: https://dataset.litnet.ltNetML-2020: https://github.com/ACANETS

## Figures and Tables

**Figure 1 sensors-20-04583-f001:**
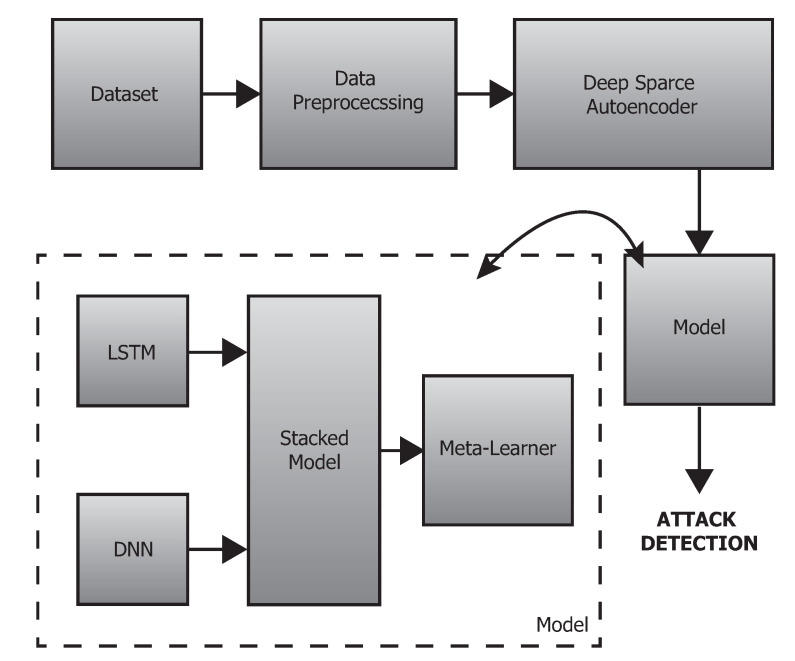
The scheme of the proposed anomaly detection mechanism.

**Figure 2 sensors-20-04583-f002:**
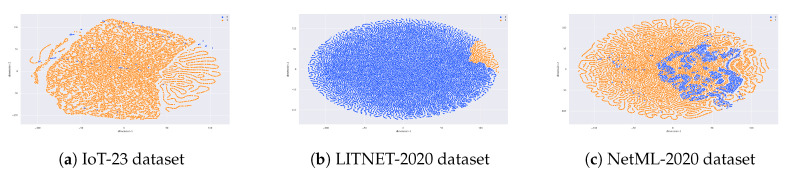
The figures depict the Deep Sparse AutoEncoder (DSAE) reconstruction procedure side-by-side with the bottleneck representation of the samples on three benchmark datasets. The *x*-axis and *y*-axis represent 1 dimension vs. 2 and the colors stand for attack classes (blue—benign, orange—malicious).

**Figure 3 sensors-20-04583-f003:**
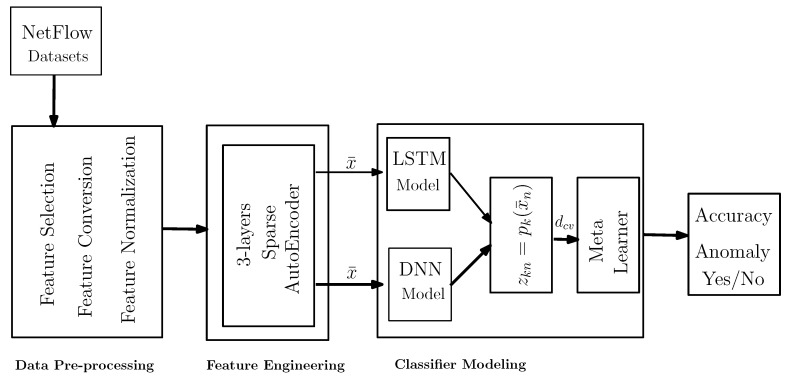
Proposed procedure of constructing a stacked ensemble classifier.

**Figure 4 sensors-20-04583-f004:**
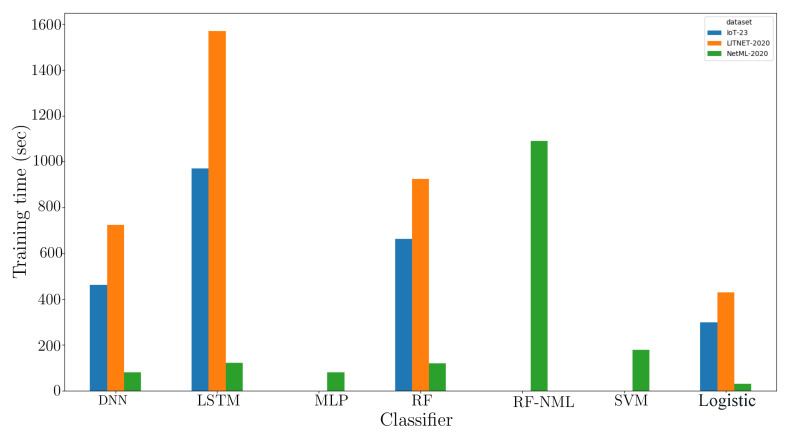
Training time taken by the proposed model vs. baseline models (RF, RF-NTML, SVM, MLP, DNN, and LSTM).

**Table 1 sensors-20-04583-t001:** Results of classifiers’ performance in terms of 5-fold cross-validation; best results are indicated with bold.

Fold	DNN	LSTM	Stacked (Proposed)
fold1	**99.97%**	99.95%	99.96%
fold2	99.92%	99.94%	**99.97%**
fold3	98.65%	99.39%	**99.96%**
fold4	99.97%	**99.98%**	**99.98%**
fold5	96.93%	99.96%	**99.99%**
Average Accuracy	99.088%	99.844%	**99.972%**
Standard Deviation	1.331	0.254	**0.013**
SEM	0.595	0.113	**0.005**

**Table 2 sensors-20-04583-t002:** Results of classifiers’ performance in terms of 5-fold cross-validation; best results are indicated with bold.

Fold	DNN	LSTM	Stacked (Proposed)
fold1	**100%**	**100%**	**100%**
fold2	99.98%	**100%**	**100%**
fold3	**100%**	99.96%	**100%**
fold4	**100%**	**100%**	**100%**
fold5	99.99%	**100%**	**100%**
Average Accuracy	99.994%	99.992%	**100%**
Standard Deviation	0.007	0.016	**0.000**
SEM	0.006	0.007	**0.000**

**Table 3 sensors-20-04583-t003:** Results of classifiers’ performance in terms of 5-fold cross-validation; best results are indicated with bold.

Fold	DNN	LSTM	Stacked (Proposed)
fold1	**99.97%**	99.95%	**99.97%**
fold2	99.92%	99.95%	**99.98%**
fold3	98.72%	99.60%	**99.99%**
fold4	99.97%	99.98%	**99.99%**
fold5	96.91%	99.96%	**99.99%**
Average Accuracy	99.098%	99.888%	**99.984%**
Standard Deviation	1.193	0.144	**0.007**
SEM	0.533	0.064	**0.003**

**Table 4 sensors-20-04583-t004:** IoT-23 datasets (capture names, malware types, and sizes).

Capture Name	Malware Name	Capture Size
Honeypot-4-1	Bening	64 k
Honeypot-5-1	Bening	180 k
Honeypot-7-1	Bening	20 k
Malware-34-1	Mirai	2.9 M
Malware-43-1	Mirai	2.9 M
Malware-1-1	Hide&Seek	2.9 M
Malware-3-1	Muhstik	8.8 G
Malware-35-1	Mirai	142 M
Malware-39-1	IRCBot	24 M
Malware-7-1	Mirai	1.3 G
Malware-8-1	Hakai	11 G
Malware-9-1	Hajime	446 M
Malware-20-1	Torli	1.4 M
Malware-21-1	Torli	948 M
Malware-42-1	Torjan	412 K
Malware-17-1	Kenjiro	420 k
Malware-36-1	Okiru	567 k
Malware-33-1	Kenjiro	7.6 G
Malware-48-1	Mirai	1.6 G
Malware-44-1	Mirai	7.4 G
Malware-49-1	Mirai	508 M
Malware-52-1	Mirai	2.9 G
Malware-60-1	Gagfyt	446 M

**Table 5 sensors-20-04583-t005:** IoT-23 dataset features’ list with descriptions.

Attribute Number	Features	Description
1	fields-ts	Flow start time
2	uid	Unique ID
3	id.orig-h	Source IP address
4	id.orig-p	Source port
5	id.resp-h	Destination IP address
6	id.resp-p	Destination port
7	proto	Transaction protocol
8	service	http, ftp, smtp, ssh, dns, etc.
9	duration	Record total duration
10	orig-bytes	Source2destination transaction bytes
11	resp-bytes	Destination2source transaction bytes
12	conn-state	Connection state
13	local-orig	Source local address
14	local-resp	Destination local address
15	missed-bytes	Missing bytes during transaction
16	history orig-pkts	History of source packets
17	orig-ip-bytes	Flow of source bytes
18	resp-pkts	Destination packets
19	resp-ip-bytes	Flow of destination bytes
20	tunnel-parents	Traffic tunnel
21	label	Attack label

**Table 6 sensors-20-04583-t006:** Sample counts for the type of attacks, number of flows, and number of attacks in the dataset.

Attack Type	Flows	Attacks
Smurf	3,994,426	59,479
ICMP-flood	3,863,655	11,628
UDP-flood	606,814	59,479
TCP SYN-flood	14,608,678	3,725,838
HTTP-flood	3,963,168	22,959
LANDattack	3,569,838	52,417
Blaster worm	2,858,573	24,291
Code red worm	5,082,952	1,255,702
Spam bot’s detection	1,153,020	747
Reaper worm	4,377,656	1176
Scanning/spread	6687	6232
Packet fragmentation attack	1,244,866	477

**Table 7 sensors-20-04583-t007:** LITNET-2020 dataset features’ list with descriptions.

Attribute Number	Features	Description
1	ts	Flow start time
..	..	*49 attributes that are specified*
..	..	*in NetFlow v9 [30]*
..	..	*+15 extended attributes*
64	tr	Flow received time-stamp
65	icmp-smf	Flooding network broadcast
66	icmp-f	Flooding target with ICMP packets
67	udp-f	Ddos’ing with UDP traffic
68	tcp-f-s	Flooding attack with SYN packets
69	tcp-f-n-a	Flooding attack with SYN packets
70	tcp-f-n-f	Flooding attack with SYN packets
71	tcp-f-n-r	Flooding attack with SYN packets
72	tcp-f-n-p	Flooding attack with SYN packets
73	tcp-f-n-u	Flooding attack with SYN packets
74	tcp-dst-p	Ddos’ing with HTTP traffic
75	tcp-land	Landing type of attack
76	tcp-src-tftp	Flooding TFTPservice
77	tcp-src-kerb	Flow of destination bytes
78	tcp-src-rpc	Flooding Kerberos service
79	tcp-dst-p-src	Flooding RPCservice
80	smtp-dst	Uses a vulnerability in an HTTP server
81	udp-p-r-range	Flooding with SMTP connections
82	p-range-dst	Scans on UDP ports 80, 8080, 81, etc.
83	udp-src-p-0	Several ports, one address
84	Label	Attack label

**Table 8 sensors-20-04583-t008:** Files used to create the NetML dataset.

Type of Attack	File Name
Bening	2013-12-17-capture1.pcap
Bening	2017-04-18-win-normal.pcap
Bening	2017-04-19-win-normal.pcap
Bening	2017-04-25-win-normal.pcap
Bening	2017-04-28-normal.pcap
Bening	2017-04-30-normal.pcap
Bening	2017-04-30-win-normal.pcap
Bening	2017-05-01-normal.pcap
Bening	2017-05-02-kali-normal.pcap
Adload	2018-05-03-win12.pcap
Artemis	capture-win15.pcap
BitCoinMiner	2018-04-04-win16.pcap
CCleaner	2017-12-18-win2.pcap
CCleaner	2018-01-30-win17.pcap
Cobalt	2018-04-03-win11.pcap
Downware	2018-02-23-win10.pcap
Dridex	2018-04-03-win12.pcap
Emotet	2017-06-24-win3.pcap
HTBot	2018-04-04-win20.pcap
MagicHound	2017-11-22-win4.pcap
MinerTrojan	2018-03-27-win4.pcap
PUA	2018-02-16-win8.pcap
PUA	2018-02-23-win11.pcap
Ramnit	2018-04-03-win6.pcap
Sality	2017-11-23-win16.pcap
Tinba	capture-win1.pcap
TrickBot	capture-win11.pcap
Trickster	2018-01-29-win7.pcap
TrojanDownloader	2018-03-27-win23.pcap
Ursnif	capture-win12.pcap
WebCompanion	2018-03-01-win9.pcap

**Table 9 sensors-20-04583-t009:** NetML-2020 dataset features’ list with descriptions.

Attribute Number	Features	Description
1	sa	source address
2	da	destination address
3	pr	protocol (6 or 17)
4	src-port	source port
5	dst-port	destination port
6	bytes-out	total bytes out
7	num-pkts-out	total packets out
8	bytes-in	total bytes in
9	time-start	time-stamp of first packet
10	time-end	time-stamp of last packet
11	intervals-ccnt[]	compact histogram of pktarriving intervals
12	ack-psh-rst-syn-fin-cnt[]	histogram of tcpflag counting
13	hdr-distinct	distinct values of header lengths
14	hdr-ccnt[]	compact histogram of header lengths
15	pld-distinct	distinct values of payload length
16	pld-ccnt[]	compact histogram of payload lengths
17	hdr-mean	mean value of header lengths
18	hdr-bin-40	pkts with header lengths between 28 and 40
19	pld-bin-128	pkts whose payload lengths are below 128
20	pld-bin-inf	pkts whose payload lengths are above 1024
21	pld-max	max value of payload length
22	pld-mean	mean value of payload length
23	pld-medium	medium value of payload length
24	pld-var	variance value of payload length
25	rev	flow features of the reverse flow
26	Label	attack label

**Table 10 sensors-20-04583-t010:** Performance evaluation of the IoT-23 test set; best results are indicated with bold. g-mean, geometric mean.

Method	Accuracy	g-Mean	Friedman Rank	Statistical Test
Random Forest [27]	0.893	–	–	No
DNN	0.984	0.932	3	Yes
LSTM	0.991	0.962	2	Yes
Stacked (proposed)	**0.997**	**0.971**	**1**	Yes

**Table 11 sensors-20-04583-t011:** Result of evaluation matrices w.r.t. existing approaches on the IoT-23 test set; best results are indicated with bold.

Method	Feature Engg.	Pr	Re	FPR	F1-Score	MCC
Random Forest [27]	AE	0.92	0.89	1.22	0.896	0.87
DNN	DSAE	**1.00**	0.89	0.23	0.87	0.98
LSTM	DSAE	**1.00**	0.92	0.19	0.95	0.99
Stacked (proposed)	DSAE	**1.00**	**0.95**	**0.13**	**0.98**	**0.99**

**Table 12 sensors-20-04583-t012:** Performance evaluation of the LITNET-2020 test set; best results are indicated with bold.

Method	Accuracy	g-Mean	Friedman Rank	Statistical Test
Random Forest [27]	0.911	–	–	No
DNN	0.997	0.991	**1**	Yes
LSTM	0.991	0.991	**1**	Yes
Stacked (proposed)	**1.00**	**0.999**	**1**	Yes

**Table 13 sensors-20-04583-t013:** Result of evaluation matrices w.r.t. existing approaches on the LITNET-2020 test set; best results are indicated with bold.

Method	Feature Engg.	Pr	Re	FPR	F1-Score	MCC
Random Forest [27]	AE	0.93	0.90	0.12	0.95	0.98
DNN	DSAE	**1.00**	**1.00**	0.1	**1.00**	0.99
LSTM	DSAE	**1.00**	**1.00**	0.07	**1.00**	0.99
Stacked (proposed)	DSAE	**1.00**	**1.00**	**0.05**	**1.00**	**1.00**

**Table 14 sensors-20-04583-t014:** Performance evaluation of the NetML-2020 test set; best results are indicated with bold.

Method	Accuracy	g-Mean	Friedman Rank	Statistical Test
Random Forest [27]	0.961	–	–	No
Random Forest [32]	0.993	–	–	No
SVM [32]	0.962	–	–	No
MLP [32]	0.988	–	–	No
DNN	0.998	0.992	2	Yes
LSTM	0.999	0.995	2	Yes
Stacked (proposed)	**1.00**	**0.999**	**1**	Yes

**Table 15 sensors-20-04583-t015:** Result of evaluation matrices w.r.t. existing approaches on the NetML-2020 test set; best results are indicated with bold.

Method	Feature Engg.	Pr	Re	FPR	F1-Score	MCC
Random Forest [27]	AE	0.98	0.987	0.3	0.98	0.92
Random Forest [32]	–	0.993	0.995	0.15	0.993	0.98
SVM [32]	–	0.98	0.982	0.42	0.98	0.92
MLP [32]	–	0.981	0.991	0.2	0.981	0.97
DNN	DSAE	0.96	0.98	0.1	**1.00**	0.99
LSTM	DSAE	**1.00**	0.989	0.21	**1.00**	0.99
Stacked (proposed)	DSAE	0.99	**0.999**	**0.0**	**1.00**	**1.00**

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
