# Peer review of "A Deep Learning Ensemble for Network Anomaly and Cyber-Attack Detection"

_sensors, 2020, doi:10.3390/s20164583_

Round 1

Reviewer 1 Report

The paper describes stacked ensemble method of classification hostile traffic in net-flow data.  Proposed by authors methods consist of LSTM and DNN neural networks and additional second layer which combines results from two neural networks. During experiments conducted, authors used three big data-sets, what should be emphasized all three are provided to the public in the 2020 year. This is big advantage of presented research. The paper is well written and poses correct structure. The only remark concerns section 3.5 where second layer, where stacked classifier is described should be improved. Authors should add more details and explain some descriptions presented in the figure 3, for example, “5f CV Holdout”. Additionally, the most in the right box in the figure is partially missing. In section 4.1 authors arbitrary use only part of data set – some remark is needed. Moreover, I have some doubts if section 5.1 which describes evaluation metrics, for example, accuracy, precision etc.,  is needed. These well-known metrics are described almost in the each paper from this field.

In the paper are some minor editorial shortcomings listed below:

  • Line 46, first NB-IoT is mentioned and later Internet of Things – should be swapped,
  • Line 140/141 – authors mentioned own research concerning mail analysis – reference will be beneficial,
  • Line 240-243 this sentence is to long and unclear – should be rewritten,
  • In line 280 – appear Network Intrusion Detection System, however in the line 66 Intrusion Detection Systems and its abbreviation is introduced,
  • In line 283 footnotes with webpages describing used data-set are introduced, they are repeated at the beginning of following three sections (line 287, line 307 and line 321) which in effect introduces some inconsistencies – in page appear footnote number but it is missing at the bottom of the page,
  • Line 302 – footnote no 4 is introduced but is missing (or empty) at the bottom of the page,
  • In most cases authors inserts ‘,’ between thousands and millions however there are some inconsistencies at the lines 297, 304, 311,
  • Line 322 appears footnote no 1 which is missing,
  • Tables 7 – 13 remarks with asterisk use to big font, in my opinion should be placed in the table caption,
  • Figure 4 – caption box in plot is to small should be resized,
  • Figure 5 – plots a) and c) ends with vertical line to 0, which probably is mistake,
  • Figure 6 – caption box in plot is to small should be resized.

Reviewer 2 Report

The paper presents an interesting proposal to ensemble learning methods using stacked generalization. The method is verified against 3 datasets. Presented approach is clear and the results obtained are comprehensive and prove the theses posed. The paper provides good contribution to the domain of ML – based malware detection based on supervised learning.

Section 1.2 clearly presents the contributions of the paper and the organization of the paper which is also clear and logical. Related work is limited to the methods investigated by the authors but is satisfactory.

However the Introduction section is very loosely related to the rest of the paper. Within the citations given in section 1 (not 1.1 and 1.2) four of them (out of seven) are self-citations. At least 3 of them related to critical infrastructure protection ([3][4][6]) could be easily omitted. The method does not seem to be targeting specifically the ICSs.

The paper has however several shortcomings, which can be overcome by minor article modifications. They have been listed below:

  1. The abstract does not express well the main focus of the paper. 4 first sentences of the abstract are loosely coupled with the following ones. In particular:

Line 1-2: "In recent years, several network intrusion detection tasks have run autonomously, without human intervention [1]."

What does it mean? It does not relate to the article.

Line 2-3: "The designed expert systems and applied machine learning algorithms are widely used to automate the recovery process"

Recovery process of what?

Line 3-7: "The interaction among various industrial control systems and the Internet environment using IoT technology makes them susceptible to both physical and cyber-attacks. Given the existence of enormous network traffic and critical Cyber-Physical Systems (CPSs), traditional methods of machine learning implemented in network anomaly detection are inefficient [2]."

It should be clearly stated in the abstract if the method is related to the IoT data, ICSs etc.

  1. The paper should be carefully read. Abbreviations should be explained (e.g. PMU devices). There are some unclear sentences e.g.:

- Line 35-36: "On the computation side, the malicious user can attack the server machines where data is stored and analytics are performed, regardless if the hosting is on-site or in a cloud platform.

What analytics? Who is doing those analytics?"

- Line 152-153: "Cerqueira et al. [22] investigate a method not unlike Stacking, where several  models e.g., [..]."

“not unlike” could be just “similar to”?

  1. Algorithm 1: It is not clear where the test dataset is used.

  1. Lines 419-422: "The main conclusion taken from these visualizations is how the gradient flow through a stacked ensemble framework is important, since it helps the network to better separate data in these anomaly spaces, allowing good anomaly detection performance even when the underlying networks are not good at identifying a specific type of anomaly."

Please compare figures 6 a/b/c for respective datasets w.r.t. these conclusions.

Reviewer 3 Report

This paper considers a novel deep learning approach to detecting network traffic anomalies in computer networks. The authors propose a reasonably complicated deep learning architecture to do preprocessing and anomaly detection, and then show through a series of tests on several datasets that their methods outperform several known basic methods, as well as outperform all of their own system's individual core components.

I did not find it particularly surprising that such a complex model could outperform many simpler ones; what is less obvious is why the novel model's training times were generally lower than some of its component pieces. I think it is important that the authors explain somewhat more about their training time methodology. All they say is that their method considered the "optimal number of features," but they do not apparently tell us whether it was a fair fight. Were the other models optimized as well, and if so, how?

Altogether the paper seems to make a meaningful contribution, and it is reasonably well written.
